

# Prognostic model of invasive ductal carcinoma of the breast based on differentially expressed glycolysis-related genes

Xiaoping Li[1,*], Qihe Yu[2,*], Jishang Chen[3], Hui Huang[4], Zhuangsheng Liu[5], Chengxing Wang[1], Yaoming He[1], Xin Zhang[6], Weiwen Li[7], Chao Li[1], Jinglin Zhao[1] and Wansheng Long[5]

[1] Department of Gastrointestinal Surgery, Affiliated Jiangmen Hospital of Sun Yat-sen University, Jiangmen, Guangdong, China
[2] Department of Oncology, Affiliated Jiangmen Hospital of Sun Yat-sen University, Jiangmen, Guangdong, China
[3] Department of Breast Surgery, Yangjiang people's Hospital, Yangjiang, Guangdong, China
[4] Department of Breast Surgery, Jiangmen Maternity & Chile Health Care Hospital, Jiangmen, Guangdong, China
[5] Department of Radiology, Affiliated Jiangmen Hospital of Sun Yat-sen University, Jiangmen, Guangdong, China
[6] Clinical Experimental Center, Jiangmen Key Laboratory of Clinical Biobanks and Translational Research, Affiliated Jiangmen Hospital of Sun Yat-sen University, Jiangmen, Guangdong, China
[7] Department of Breast and Thyroid Surgery, Affiliated Jiangmen Hospital of Sun Yat-sen University, Jiangmen, Guangdong, China
* These authors contributed equally to this work.

Corresponding author
Wansheng Long, jmlws2@163.com

## ABSTRACT

**Background**. Invasive ductal carcinoma (IDC) is a common pathological type of breast cancer that is characterized by high malignancy and rapid progression. Upregulation of glycolysis is a hallmark of tumor growth, and correlates with the progression of breast cancer. We aimed to establish a model to predict the prognosis of patients with breast IDC based on differentially expressed glycolysis-related genes (DEGRGs).

**Methods**. Transcriptome data and clinical data of patients with breast IDC were from The Cancer Genome Atlas (TCGA). Glycolysis-related gene sets and pathways were from the Molecular Signatures Database (MSigDB). DEGRGs were identified by comparison of tumor tissues and adjacent normal tissues. Univariate Cox regression and least absolute shrinkage and selection operator (LASSO) regression were used to screen for DEGRGs with prognostic value. A risk-scoring model based on DEGRGs related to prognosis was constructed. Receiver operating characteristic (ROC) analysis and calculation of the area under the curve (AUC) were used to evaluate the performance of the model. The model was verified in different clinical subgroups using an external dataset (GSE131769). A nomogram that included clinical indicators and risk scores was established. Gene function enrichment analysis was performed, and a protein-protein interaction network was developed.

**Results**. We analyzed data from 772 tumors and 88 adjacent normal tissues from the TCGA database and identified 286 glycolysis-related genes from the MSigDB. There were 185 DEGRGs. Univariate Cox regression and LASSO regression indicated that 13 of these genes were related to prognosis. A risk-scoring model based on these

13 DEGRGs allowed classification of patients as high-risk or low-risk according to median score. The duration of overall survival (OS) was longer in the low-risk group ($P < 0.001$), and the AUC was 0.755 for 3-year OS and 0.726 for 5-year OS. The results were similar when using the GEO data set for external validation (AUC for 3-year OS: 0.731, AUC for 5-year OS: 0.728). Subgroup analysis showed there were significant differences in OS among high-risk and low-risk patients in different subgroups (T1-2, T3-4, N0, N1-3, M0, TNBC, non-TNBC; all $P < 0.01$). The C-index was 0.824, and the AUC was 0.842 for 3-year OS and 0.808 for 5-year OS from the nomogram. Functional enrichment analysis demonstrated the DEGRGs were mainly involved in regulating biological functions.

**Conclusions**. Our prognostic model, based on 13 DEGRGs, had excellent performance in predicting the survival of patients with IDC of the breast. These DEGRGs appear to have important biological functions in the progression of this cancer.

## INTRODUCTION

Breast invasive ductal carcinoma (IDC) is the most common malignant tumor in females worldwide (*Harbeck et al., 2019*; *Hanker, Sudhan & Arteaga, 2020*). In 2018, there were more than 266,000 cases of breast IDC among females in the United States, and this cancer accounted for 30% of malignant tumors in females, far more than lung cancer (13%) (*Bray et al., 2018*; *Ahmad, 2019*). The prognosis of women with breast IDC is related to the activation or silencing of various biological functions in tumor tissues and signaling pathways. There are prognostic models based on tumor immunity and autophagy, but no models have exclusively focused on IDC (*Li et al., 2020a*; *Li et al., 2020b*; *Zhang, Zhang & Yu, 2019*; *Hu et al., 2020*) and few models examined genes that function in basic metabolism.

Glycolysis is a series of reactions that catabolize most carbohydrates. "Metabolic reprogramming" is the hallmark of tumors, and glycolysis is the main source of energy for tumor cells, even when there is insufficient oxygen (*Wu et al., 2020*). Moreover, activation of glycolysis-related genes occurs in almost all tumor cells. For example, *Dai et al. (2020)* found that glycolysis promoted the progression of pancreatic cancer and induced gemcitabine chemotherapy resistance. Long noncoding RNAs (lncRNAs) that interact with Long Intergenic Noncoding RNA for IGF2BP2 Stability (LINRIS) activate aerobic glycolysis in tumor cells, and can affect the development and prognosis of rectal cancer (*Wang et al., 2019a*; *Wang et al., 2019b*).

Research on the function of glycolysis-related genes in breast tumors showed that hexokinase (HK2) had high expression in breast IDC cells, and that HK2 silencing inhibited IDC (*Patra et al., 2013*; *Cao et al., 2020*). Other research showed that overexpression of 6-Phosphofructo 2-kinase/fructose 2, 6-bisphosphatase 3 (PFKFB3) promoted the progression of breast IDC, and had negative associations with progression-free survival

(PFS), distant metastasis-free survival (DMFS), and overall survival (OS) in patients with breast IDC (*O'Neal et al., 2016*; *Peng et al., 2018a*; *Peng et al., 2018b*). Thus, glycolysis-related genes have a potentially significant impact on the progression of breast tumors and on the survival and prognosis of patients with breast tumors.

Our aim was to develop a prognostic model of breast IDC based on glycolysis-related genes and determine the potential functions of glycolysis-related genes in the progression of breast IDC.

## MATERIALS AND METHODS

### Data resources and preprocessing

The transcriptome data and corresponding clinical data of breast invasive ductal carcinoma were downloaded from the TCGA database (https://www.cancer.gov/) (*Tomczak, Czerwinska & Wiznerowicz, 2015*). The data set of glycolysis-related genes was obtained from the MSigDB database (http://www.hmdb.ca). Using $|\log_2 FC| > 0.5$ and false discovery rate (FDR) <0.05 as the cut-off value, the data was normalized with the "edgeR" package from R, and then the differential analysis was performed to obtain the differential expression of glycolysis-related genes (DEGRGs) between the tumor tissue and normal tissue.

### Construction of risk-scoring model

Based on the above DEGRGs, univariate Cox regression and LASSO regression were used to screen out prognostic-related glycolysis genes. The risk score was evaluated by the coefficient of each prognostic-related glycolysis gene. The risk scoring formula was constructed as Risk scores $= \sum_1^i (\text{coefi}^\star \text{expri})$, where $i$ is the number of genes used to build the model, coefi is the coefficients of the genes in the model, and expri the expression of genes in the model. Taking the median risk score as the cut-off point, patients were divided into high-risk and low-risk groups. The survival outcome of the two groups was observed by Kaplan–Meier survival analysis. Receiver operating characteristic (ROC) curve was applied to calculate the area under the curve (AUC) to evaluate the predictive ability of the risk-scoring model. Independent GEO (https://www.ncbi.nlm.nih.gov/geo/) data sets are used to verify the above results (*Barrett et al., 2013*). Univariate Cox regression and multivariate Cox regression were used to identified the independent prognostic factors among risk scores, age, tumor TNM stage, and whether triple negative breast cancer (TNBC) or not. Through clinical survival analysis, the predictive ability of the risk-scoring model in different clinical subgroups was clarified.

### External validation of the risk scoring model

The TCGA results were validated using the GEO (https://www.ncbi.nlm.nih.gov/geo/) dataset (GSE131769). For this validation, the outcomes of the two groups were compared using Kaplan–Meier survival analysis. ROC curves were used to calculate AUCs and evaluate the predictive performance of the risk-scoring model.

## Construction of the nomogram

A nomogram was constructed based on the results of the multivariate Cox regression, with clinical information such as age, TMN stage, TNBC status, and DEGRG risk scores, to predict 3-year and 5-year OS. The predictive ability of the nomogram was evaluated by calculating the C-index and the calibration chart, clinical decision curve analysis, and an ROC curve.

## Function enrichment analysis

We analyzed the genes that were differentially expressed in the high-risk and low-risk groups using the Kyoto Encyclopedia of Genes and Genomes (KEGG) and Gene Ontology (GO) to identify pathway enrichment (*Kanehisa et al., 2019*). This analysis allowed identification of the functions of the differentially expressed genes. We then examined whether the differentially expressed genes were involved in the development of breast cancer.

## Construction of interactive network diagram

To determine the relationship of the DEGRGs model with prognosis, a network between genes was developed. The prognosis-related genes were imported into Search Tool for the Retrieval of Interacting Genes Proteins (STRING) to construct an interaction network (*Szklarczyk et al., 2019*).

# RESULTS

## DEGRGs in breast IDC

We obtained gene expression data and clinical data of females with IDC of the breast (772 tumor tissues, 88 adjacent normal tissues) from the TCGA database and the glycolysis gene set (286 genes) from the GSEA website. Based on standard cut-off values for fold-change in gene expression ($|\log_2(\text{FC})| > 0.5$) and false discovery rate (FDR < 0.05), the IDC tissues had 185 DEGRGs, with 67 down-regulated genes and 118 up-regulated genes (Figs. 1A, 1B, Table S1).

## Relationship of DEGRGs with prognosis and risk-scoring model

We then identified patients with follow-up times greater than 30 days, and performed univariate Cox regression and LASSO regression to screen for DEGRGs that were related to prognosis (Figs. 2A, 2B, 2C). This analysis indicated that 13 DEGRGs were closely related to prognosis.

We constructed a risk-scoring model based on multivariate Cox regression and divided patients into high-risk and low-risk groups based on median risk score. Kaplan–Meier survival analysis showed that patients with high-risk had significantly reduced duration of OS ($P = 9.795 \times 10^{-8}$, Fig. 3A). ROC analysis indicated the AUC was 0.755 for 3-year OS and 0.726 for 5-year OS (Fig. 3B). The risk curve and scatterplot (Figs. 3C, 3D) show the risk scores and survival status of all patients, and indicate that the risk coefficient and mortality rate were greater in the high-risk group. We plotted a "heat map" to visualize the expression of the 13 DEGRGs in the high-risk and low-risk groups (Fig. 3E). Taken together, these results confirm that 13 DEGRGs were significant prognostic indicators for patients with IDC of the breast.

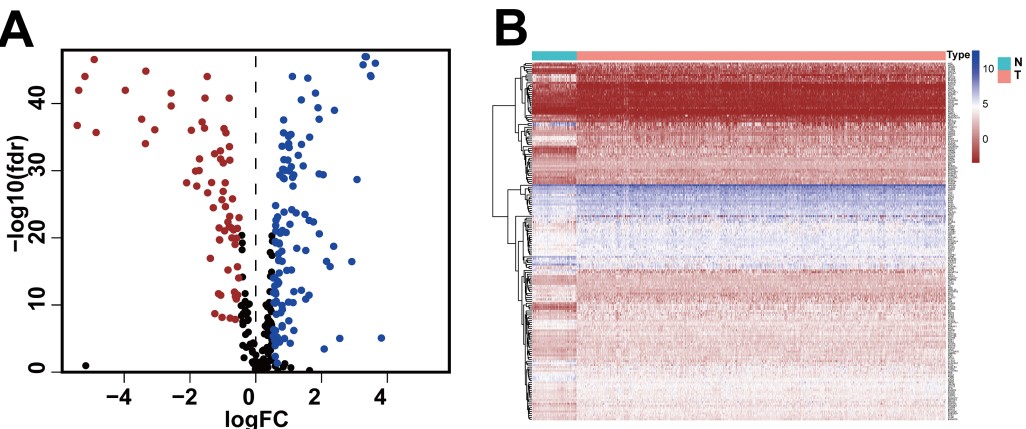

**Figure 1  Differentially expressed glycolysis-related genes between breast invasive ductal carcinoma and normal tissues.** (A) The volcano gram showed that compared with normal tissues, 67 DEGRGs were down-regulated and 118 DEGRGs were up-regulated in breast invasive ductal carcinoma. ($P < 0.05$) (B) The heat map showed the expression of 185 DEGRGs in both tumor tissues and normal tissues.

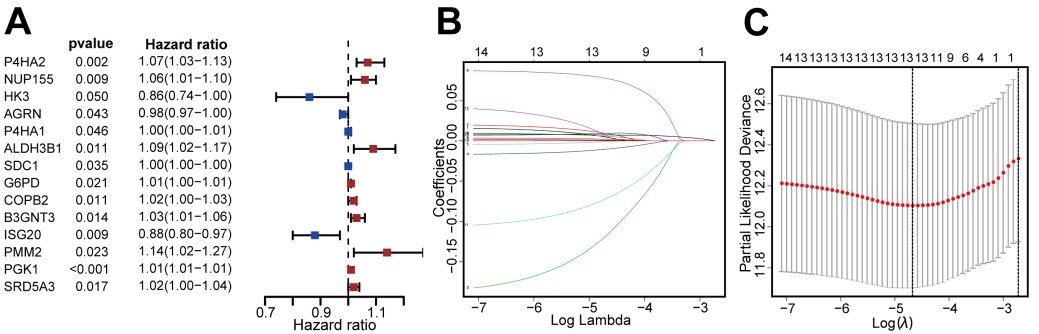

**Figure 2  Screening DEGRGs with prognostic value.** (A) Based on univariate Cox regression, the forest map showed that there were 14 DEGRGs with prognostic significance ($P < 0.05$). (B, C) LASSO regression further screened out 13 DEGRGs that were closely related to the prognosis.

We performed univariate and multivariate Cox regression to evaluate the effect of risk score, age, triple-negative breast cancer (TNBC), and TNM stage on patient prognosis (Figs. 4A, 4B). The results confirmed that the risk score was an independent prognostic factor for patients with IDC of the breast (adjusted hazard ratio: 2.71, 95% CI [1.87–3.94]).

We also performed survival analysis of different subgroups based on TNM status (Fig. 5). This analysis indicated that the risk-scoring model had good predictive value in the T1-2 subgroup, T3-4 subgroup, N0 subgroup, N1-3 subgroup, M0 subgroup, TNBC subgroup, and non-TNBC subgroup (all $P < 0.001$), but not in the M1 subgroup ($P = 0.857$).

### External validation of the risk scoring model

We verified the model using the GEO dataset (GSE131769). Kaplan–Meier survival curves showed that patients with high-risk had a significantly shorter duration of OS ($P = 3.245$

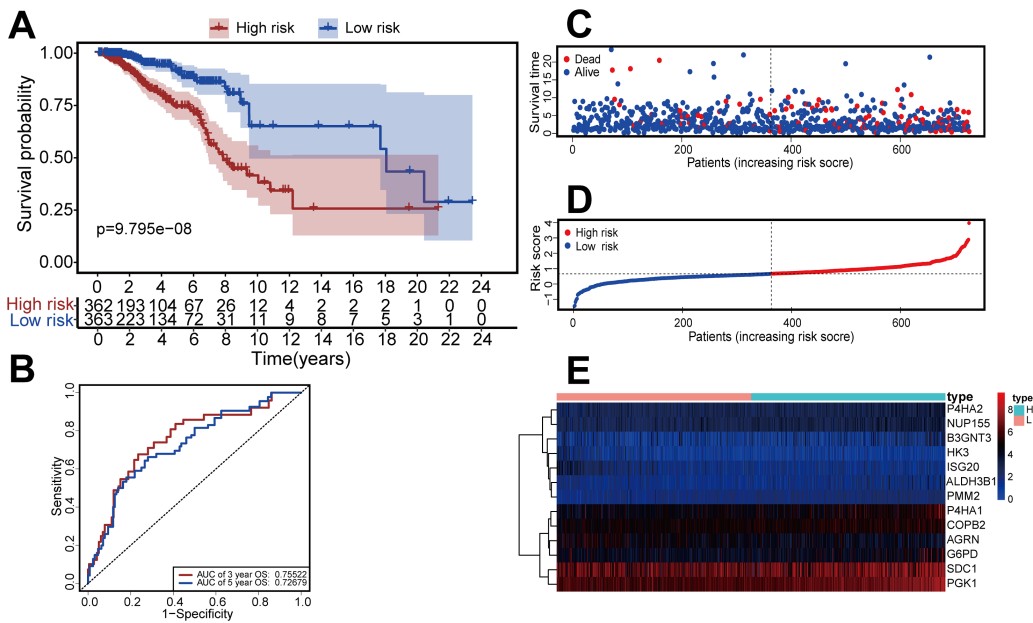

**Figure 3** **Construction of the risk-scoring model for patients with breast invasive ductal carcinoma based on DGRG.** (A) Kaplan–Meier survival analysis showed that patients in the high-risk group had a shorter OS than that of the low-risk group ($P = P = 9.795 \times 10^{-8}$). (B) The ROC curve showed that the AUC of the 3-year OS and 5-year OS were 0.755 and 0.726, respectively. (C–E) The overview of survival time for each patient, the distributions of risk scores for each patient and heatmaps of expression profiles for 13-DEGRGs between the low-risk group and the high-risk group.

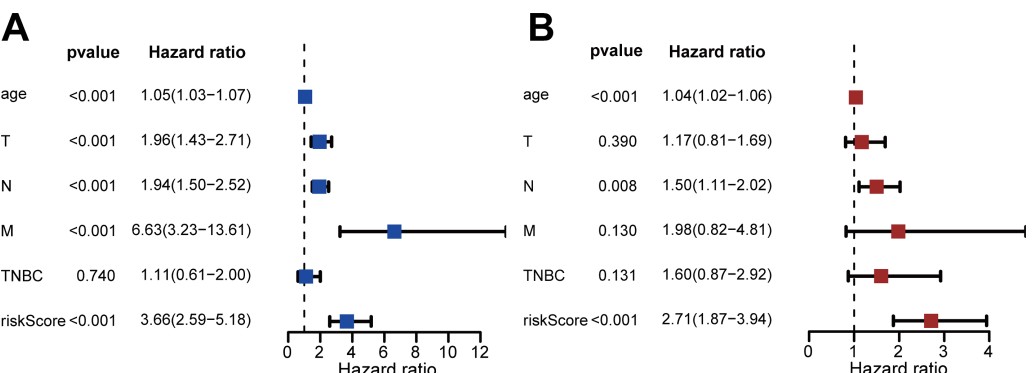

**Figure 4** **Assessment of risk scores and prognostic value of clinical data.** (A) Univariate Cox analysis showed that risk scores and clinical variables including age, TMN stage, and whether it was TNBC or not were significantly related to OS. (B) Multivariate Cox analysis manifested that the 13-DEGRGs signature was an independent prognostic indicator for patients with breast invasive ductal carcinoma.

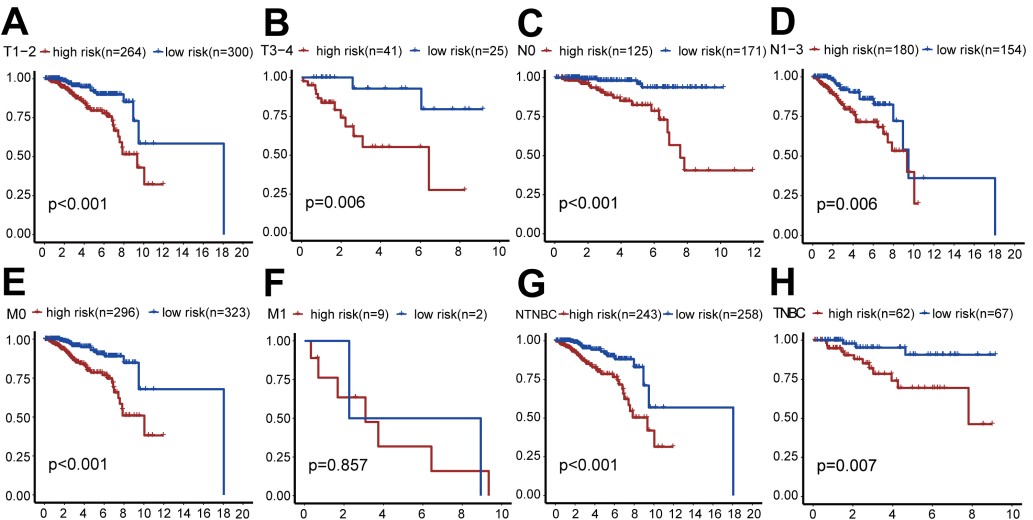

**Figure 5** Clinical survival analysis of subgroups' DGRG risk scores for breast invasive ductal carcinoma. (A) In the T1–T2 subgroup, the OS of the high-risk group was lower than that of the low-risk group ($P < 0.001$). (B) In the T3–T4 subgroup, the OS of the high-risk group was lower than that of the low-risk group ($P = 0.006$). (C) In the N0 subgroup, the OS of the high-risk group was lower than that of the low-risk group ($P < 0.001$). (D) In the N0–N3 subgroup, the OS of the high-risk group was lower than that of the low-risk group ($P = 0.006$). (E) In the M0 subgroup, the OS of the high-risk group was lower than that of the low-risk group ($P < 0.001$). (F) In the M1 subgroup, there was no significant difference in OS between patients in the high-risk group and the low-risk group, because the number of cases in the M1 subgroup is relatively small ($P = 0.857$). (G) In the TNBC subgroup, the OS of the high-risk group was lower than that of the low-risk group ($P < 0.001$). (H) In the NTNBC subgroup, the OS of the high-risk group was lower than that of the low-risk group ($P = 0.007$). The above results suggested that the DEGRGs risk-scoring model had a good predictive ability.

$\times 10^{-3}$, Fig. 6A). ROC analysis indicated that the AUC was 0.731 for 3-year OS and was 0.728 for 5-year OS (Fig. 6B). These results confirm the validity of our risk scoring model.

## Construction of the prediction model

Based on the results of the multivariate Cox regression, we developed a nomogram based on age, TMN stage, risk score, and TNBC status to predict 3-year and 5-year OS (Fig. 7A). We then used the C-index, clinical decision curve, calibration chart, and ROC curve to evaluate the predictive performance of the nomogram (Figs. 7B–7F). The results indicated that the prognostic model had good prediction accuracy, with a C-index of 0.824, an AUC for 3-year OS of 0.842, and an AUC for 5-year OS of 0.808. These results verified the predictive ability of the nomogram.

## Gene function enrichment analysis

To explore the potential mechanisms of prognosis-related DEGRGs in breast invasive ductal carcinoma, we performed KEGG enrichment analysis and GO enrichment analysis. KEGG pathway enrichment analysis showed that the gene set was enriched in the cell cycle, DNA replication, glycolysis and gluconeogenesis, RNA degradation, arachidonic acid metabolism, cytokine-cytokine receptor interactions, cytosolic DNA sensing, and ribosome function (Figs. 8A–8H). GO enrichment analysis showed that the gene set was enriched in

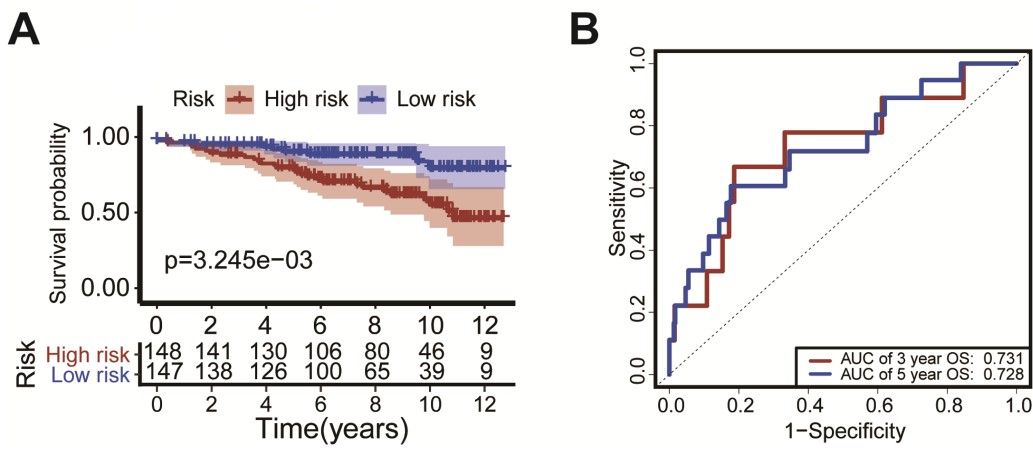

**Figure 6** **External validation of the risk scoring model The model was verified in GEO datasets** (**GSE131769**). Kaplan-Meier survival curves showed that the OS of patients with high-risk was significantly shorter than in the low-risk group ($P = P = 3.245 \times 10^{-3}$, Fig. 6A). The ROC curve was drawn to calculate the AUC of the 3-year OS and 5-year OS as 0.731 and 0.728, respectively (Fig. 6B).

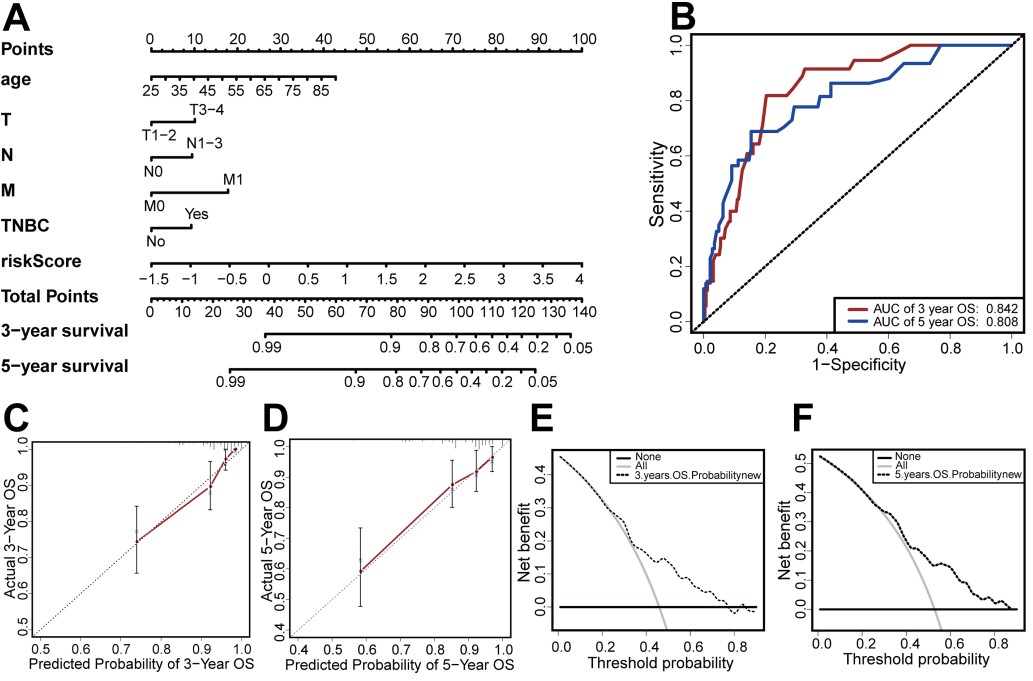

**Figure 7** **Establishment and evaluation of the nomogram.** (A) The nomograms for predicting the patients' OS. (B) ROC curve analysis showed that AUC of 3-year and 5-year OS were 0.842 and 0.808, respectively. (C) The calibration curve for the 3-year OS of the nomogram. (D) The calibration curve for the 5-year OS of the nomogram. (E) The clinical decision curve for the 3-year OS of the nomogram. (F) The clinical decision curve for the 5-year OS of the nomogram.

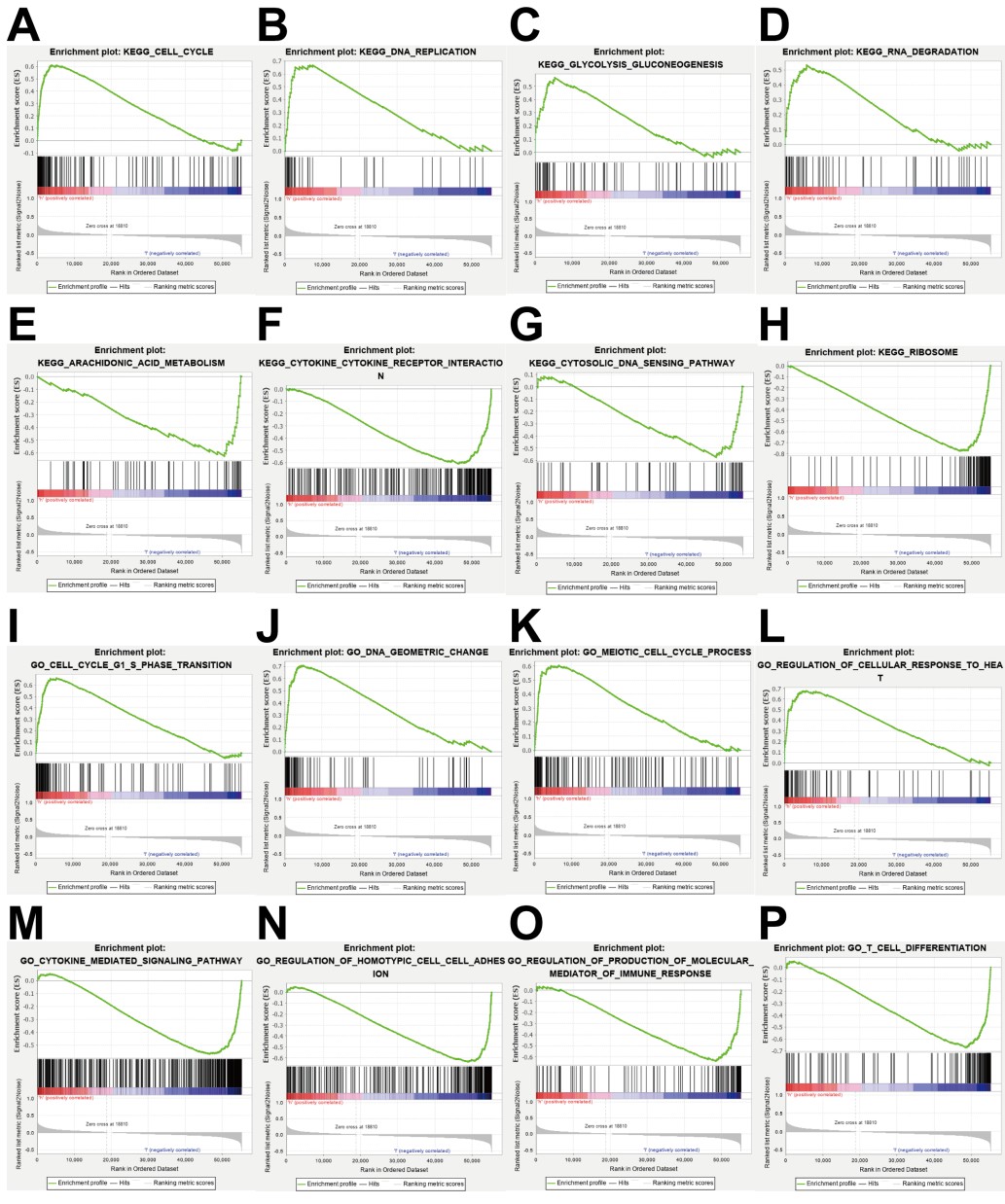

**Figure 8** **Enrichment analysis.** (A–H) KEGG pathway analysis showed that these genes were involved in cell cycle, DNA replication, glycolysis gluconeogenesis, RNA degradation, arachidonic acid metabolism, cytokine cytokine receptor interaction, cytosolic DNA sensing pathway and ribosome. (I–P) GO enrichment analysis showed that the genes were enriched in the cell cycle G1-S phase transition, DNA geometric change, meiotic cell cycle process, regulation of cellular response to heat, cytokine mediated signaling pathway, regulation of homotypic cell cell adhesion, regulation of production of molecular mediator of immune response, T cell differentiation.

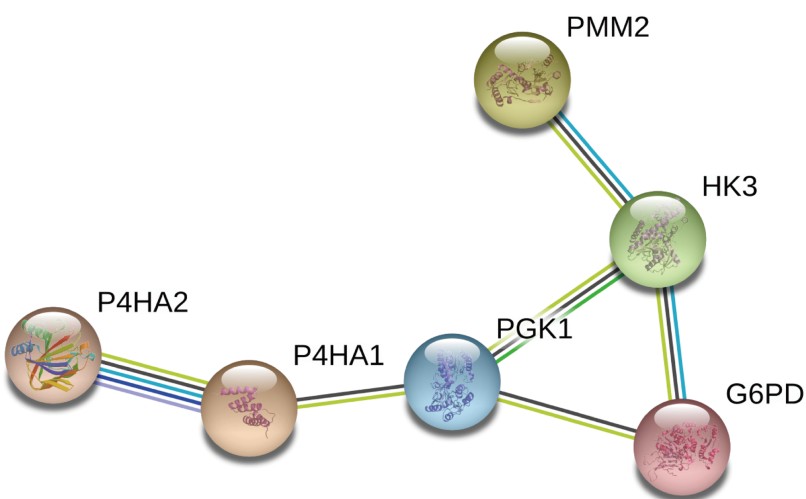

**Figure 9** **Construction of a network diagram of the interaction between DEGRGs.** (A) The analysis of STRING showed that there were interactions among six genes (P4HA2, P4HA1, PGK1, G6PD, HK3, PMM2) in DGRG.

the cell cycle G1-S phase transition, DNA geometric changes, the meiotic cell cycle process, regulation of cellular response to heat, cytokine-mediated signaling pathways, regulation of homotypic cell–cell adhesion, regulation of production of molecular mediators of immune responses, and T cell differentiation (Figs. 8I–8P).

## Construction of an interactive network diagram

We constructed a network diagram to visualize the interactions between hub genes to better understand the potential functions of the different DEGRGs on prognosis. The results from STRING showed there were 6 interacting hub genes: P4HA2, P4HA1, PGK1, G6PD, HK3, and PMM2 (Fig. 9).

## DISCUSSION

Breast IDC is the most common pathological type of breast tumor, and morbidity and mortality from this cancer continue to increase (*Harbeck et al., 2019*; *Badve & Gokmen-Polar, 2019*). There is evidence that changes in glycolysis-related genes have multiple effects on the prognosis of these patients (*Li et al., 2020a*; *Li et al., 2020b*; *Chen et al., 2019*). In particular, tumor cells reprogram the glycolysis pathway to accommodate the increased energy required for malignant transformations, including invasion and metastasis (*Shen et al., 2020*; *Abbaszadeh, Cesmeli & Biray, 2020*). Given the importance of glycolysis on tumor prognosis, numerous research groups have developed models based on glycolysis-related genes in their studies of different cancers (*Zhang, Zhang & Yu, 2019*; *Wang et al., 2019a*; *Wang et al., 2019b*; *Karasinska et al., 2020*). However, no previous study developed a prognostic model for patients with breast IDC based on glycolysis-related genes.

In this research, we identified 13 DEGRGs that were related to prognosis in patients with breast IDC, and then established a risk-scoring model. The results showed that the
OS of patients in the high-risk group was significantly shorter than that of patients in the low-risk group. We also examined the impact of patient age, TMN stage, TNBC status, and risk score to construct a nomogram that predicts the 3-year and 5-year OS of these patients. Our application of various evaluation methods indicated that the nomogram had good performance in the prediction of OS. Our ROC analysis showed that the AUC was 0.842 for 3-year OS and 0.808 for 5-year OS, higher than the AUC values reported in previous models (*Lin et al., 2020*; *Xie et al., 2019*), thus indicating that our model had better predictive ability.

Among the 13 DEGRGs we used to construct the risk model, increased levels of P4H2A, NUP155, ALDH3B1, SDC1, G6PD, COPB2, B3GNT3, PMM2, and PGK1 were associated with poor prognosis and increased levels of HK3, AGRN, P4HA1, and ISG20 were associated with favorable prognosis (Table S2). Previous research reported increased levels of P4HA2 and P4HA1 (the two isomers of collagen prolyl 4-hydroxylase) in several types of human cancers and that both enzymes promoted glycolysis in tumor cells (*Li et al., 2019*). PGK1 is the first key enzyme to produce ATP in the glycolytic pathway, PGK1 is not only a metabolic enzyme but also a protein kinase, which mediates the tumor growth, migration and invasion through phosphorylation some important substrates (*Fu & Yu, 2020*). There is also evidence that SDC1 can promote the migration of breast cancer cells across the blood–brain barrier by regulating the expression of cytokines, thus promoting brain metastasis (*Sayyad et al., 2019*). Mele et al. found that the overexpression of G6PD can induce lapatinib resistance in breast cancer and also found a significant correlation between high expression of G6PD and tumor recurrence (*Mele et al., 2019*). Sauter et al. found that the level of HK3 in the nipple aspirate of patients with breast cancer was significantly lower than that of healthy women, and considered an elevated HK3 level as a possible sign of early breast cancer (*Mannello & Gazzanelli, 2001*). Thus, these previous studies are consistent with our finding that glycolysis-related genes were closely related to the occurrence and development of breast cancer and the prognosis of patients.

To further characterize the potential roles of the 13 DEGRGs in our risk model, we performed GO and KEGG enrichment analysis. The results showed that the gene set was enriched in cell cycle, DNA replication, glycolysis gluconeogenesis, RNA degradation, arachidonic acid metabolism, cytokine cytokine receptor interaction, cytosolic DNA sensing pathway and ribosome. Previous studies showed that under hypoxic conditions, metabolic reprogramming of breast tumor stem cells helped to maintain their growth and proliferation (*Peng et al., 2018a*; *Peng et al., 2018b*). Breast cancer occurs in patients with impaired immune function, in which cytokines function as growth signals for tumor cells. Many studies showed that interactions between the immune system and cancer cells, which are mediated by cytokines and chemokines, affect the initiation and progression of breast cancer and the response to treatment (*Lim et al., 2018*; *King, Mir & Singh, 2017*; *Fabre et al., 2018*). DNA replication is a fundamental biological process, and replication disorders can lead to genomic instability, an important feature of breast cancer. Many experimental and clinical studies have identified disorders of DNA replication during the development and progression of breast cancer (*Kitao et al., 2018*). Aerobic glycolysis pathway includes hexokinase, phosphofructokinase (PFK), and other genes (*Wu et al., 2020*). These previous

findings therefore support our conclusion that the 13 glycolysis-related genes identified here play an important role in the progression of breast tumors.

Our study has some limitations. Firstly, the predictive model lacks information about patient treatments, thus limiting its predictive performance. Secondly, this study was based on bioinformatics analysis, and further studies are needed to determine the potential functional mechanisms.

## CONCLUSIONS

In conclusion, our prediction model, which is based on 13 DEGRGs and the clinical characteristics of patients, can reliably predict the OS of patients with breast IDC. These 13 DEGRGs and several related miRNAs thus appear to play an important role in the development and progression of breast IDC.

### Funding

This work was supported by the Guangdong Medical Research Foundation (Grant No. A2020622) and the Elite Young Scholars Program of Jiangmen Central Hospital (Grant No. J201905). The funders had no role in study design, data collection and analysis, decision to publish, or preparation of the manuscript.

### Grant Disclosures

The following grant information was disclosed by the authors:
Guangdong Medical Research Foundation: A2020622.
Elite Young Scholars Program of Jiangmen Central Hospital: J201905.

### Competing Interests

The authors declare there are no competing interests.

### Author Contributions

- Xiaoping Li and Qihe Yu conceived and designed the experiments, performed the experiments, analyzed the data, prepared figures and/or tables, authored or reviewed drafts of the paper, and approved the final draft.
- Jishang Chen and Hui Huang performed the experiments, prepared figures and/or tables, and approved the final draft.
- Zhuangsheng Liu performed the experiments, authored or reviewed drafts of the paper, and approved the final draft.
- Chengxing Wang, Yaoming He, Xin Zhang, Weiwen Li and Chao Li analyzed the data, authored or reviewed drafts of the paper, and approved the final draft.
- Jinglin Zhao performed the experiments, prepared figures and/or tables, authored or reviewed drafts of the paper, and approved the final draft.
- Wansheng Long conceived and designed the experiments, performed the experiments, prepared figures and/or tables, authored or reviewed drafts of the paper, and approved the final draft.

## Data Availability

The RNA-seq transcriptome data and corresponding clinicopathological information of the data are available at the TCGA project: TCGA-BRCA.

The raw measurements are available in the Supplemental Tables.

## Supplemental Information

Supplemental information for this article can be found online at http://dx.doi.org/10.7717/peerj.10249#supplemental-information.

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
