# Peer review of "Prognostic model of invasive ductal carcinoma of the breast based on differentially expressed glycolysis-related genes"

_PeerJ, doi:10.7717/peerj.10249_

## Round 0.1 · original submission · Minor Revisions

As you can see, both reviewers are mostly positive but they have several issues that need to be addressed.

Reviewer 1 ·

Basic reporting

no comment

Experimental design

no comment

Validity of the findings

no comment

Additional comments

The manuscript by Li et al constructed a model based on the RNA-seq data of glycolysis associated genes in breast cancer. Overall, this study is interesting. It provides a new model to predict the survival of IDC breast cancer. It is well designed and performed. I only have several minor concerns.
Minor points:
1. They should include the references for all the software and data they used in this this study.
2. Can they change the color of Fig. 1B to make it look brighter? The background is a little dark right now.
3. The analysis of STRING showed that there were interactions among 6 genes (P4HA2, P4HA1, PGK1, G6PD, HK3, PMM2) in DGRG. They should have some introduction of these 6 genes, so that readers can understand it better.
4. In Fig. 10, did they include all the tumor tissue or just the IDC tumor tissue?
5. They could consider combine Fig. 9 and Fig.10 together.
6. They found 185 DGRGs between tumors and normal tissues. If they can include a table as supplementary data to show these 185 DGRGs and the p values, it will be better.

Reviewer 2 ·

Basic reporting

This is an interesting manuscript by Li et al. which investigates the role of expression of glycolysis-related genes (GRGs) in predicting prognosis of breast cancer within the type of IDC. The manuscript is well written and the tables and figures are well organized.

Experimental design

The present study performed a systematic analysis of GRGs in IDC of breast cancer to explore its prognostic significance and uncover its potential function and mechanism using the database of TCGA. The analysis is classic, well described and reasonable, and the results are convinced.

Validity of the findings

Generally, the findings are meaningful and the conclusion is clearly stated based on the results. But there is still some weakness:
Major comments:
1. GO and KEGG enrichment analysis are good to reveal the underline function of the GRGs. However, the functions and networks the authors found are not directly related to the biology of breast cancer. I suggest the author could perform further KEGG analysis or GSEA analysis to find whether the GRG signature has close relationship with cancer gene sets, like immunity, proliferation, DNA damage and so on.
2. Analysis of the miRNA is not necessary. I suggest the author could delete this part.

Minor points:
1. The abbreviation of differentially expressed glycolysis-related genes is DGRGs in the manuscripts. It is better to use (DEGRGs).
2. It is a bioinformatic analysis, and the potential functional mechanisms were not studied. It is better for the author to address the limitation in the “Discussion” section.

Additional comments

1. Focusing on the role of GRGs in IDC of breast cancer is a good direction.
2. The anaylsis is quite clear and comprehensive.
3. The weakness is the part which uncover the function and mechnism of GRGs. The authors could make some further analysis, and/or explain more in detail.

---

## Round 0.2 · accepted · Accept

All critiques were adequately addressed and the manuscript was revised accordingly. Therefore I am pleased to accept the mended version.